# Diagnostic and Therapeutic Potential of TSPO Studies Regarding Neurodegenerative Diseases, Psychiatric Disorders, Alcohol Use Disorders, Traumatic Brain Injury, and Stroke: An Update

**DOI:** 10.3390/cells9040870

**Published:** 2020-04-02

**Authors:** Jasmina Dimitrova-Shumkovska, Ljupcho Krstanoski, Leo Veenman

**Affiliations:** 1Department of Experimental Biochemistry, Institute of Biology, Faculty of Natural Sciences and Mathematics, University Ss Cyril and Methodius, Arhimedova 3, P.O. Box 162, 1000 Skopje, Republic of North Macedonia; lkrstanoski@gmail.com; 2Technion-Israel Institute of Technology, Faculty of Medicine, Rappaport Institute of Medical Research, 1 Efron Street, P.O. Box 9697, Haifa 31096, Israel

**Keywords:** brain disorders, brain disease, TSPO, microglia, astrocytes, neurons, microglia activation, cell death, regeneration, PET tracing, drug development

## Abstract

Neuroinflammation and cell death are among the common symptoms of many central nervous system diseases and injuries. Neuroinflammation and programmed cell death of the various cell types in the brain appear to be part of these disorders, and characteristic for each cell type, including neurons and glia cells. Concerning the effects of 18-kDa translocator protein (TSPO) on glial activation, as well as being associated with neuronal cell death, as a response mechanism to oxidative stress, the changes of its expression assayed with the aid of TSPO-specific positron emission tomography (PET) tracers’ uptake could also offer evidence for following the pathogenesis of these disorders. This could potentially increase the number of diagnostic tests to accurately establish the stadium and development of the disease in question. Nonetheless, the differences in results regarding TSPO PET signals of first and second generations of tracers measured in patients with neurological disorders versus healthy controls indicate that we still have to understand more regarding TSPO characteristics. Expanding on investigations regarding the neuroprotective and healing effects of TSPO ligands could also contribute to a better understanding of the therapeutic potential of TSPO activity for brain damage due to brain injury and disease. Studies so far have directed attention to the effects on neurons and glia, and processes, such as death, inflammation, and regeneration. It is definitely worthwhile to drive such studies forward. From recent research it also appears that TSPO ligands, such as PK11195, Etifoxine, Emapunil, and 2-Cl-MGV-1, demonstrate the potential of targeting TSPO for treatments of brain diseases and disorders.

## 1. Introduction

The core subject of this review is the targeting of 18-kDa translocator protein (TSPO) for: (1) Diagnostic approaches, in particularspecific positron emission tomography (PET)scan) to visualize brain damage due to disease and injury; and (2) therapeutic approaches, i.e., application of TSPO ligands with different structures and properties to animal models for Parkinson disease (PD), Huntington disease, traumatic brain injury (TBI), and stroke. The point: TSPO ligands of different molecular structures and properties can ameliorate various brain diseases, injuries, and disorders.

## 2. Background

The 18-kDa translocator protein (TSPO), formerly known as peripheral benzodiazepine receptor (PBR), is abundantly expressed in many tissues [1]. Subcellular, its primary location is in the outer mitochondrial membrane (OMM), where it was originally found to participate in the binding of diazepam [1,2]. Establishing that this interaction was made possible by a binding site different from that of the central benzodiazepine receptor (CBR) underlined the unique nature of TSPO as a transmembrane receptor [1,2]. Later discoveries about its involvement in cholesterol transport over the OMM, thereby an indication of its interaction in neurosteroid synthesis, highlighted TSPO as one of the key players for preserving normal physiological functions in the Central Nervous System (CNS) (Figure 1) [2,3]. Furthermore, since mitochondria produce significant levels of reactive oxygen species (ROS) and thereby can contribute to cellular oxidative damage, TSPO is attracting attention as a possible modulator of oxidative stress, in addition to its inclusion in the list of biomarkers for different neuronal diseases [4].

In parallel, reported oscillations of TSPO expression in liver, aorta, and platelets in conditions of increased oxidative stress emphasize its ROSmodulating features and its participation in inflammatory processes [5,6]. Furthermore, TSPO’s ubiquitous expression also instigated extensive research concerning its involvement in other cellular pathways, for instance: Heme synthesis, modulation of gene expression, for coding and non-coding RNA, and modulation of the initiation of programmed cell death [7,8,9,10,11]. An emphasis is placed on TSPO’s participation in homeostasis in general, and in health maintenance and disease counteraction in particular.

In this light, alterations of TSPO expression as a mechanism for adaptation to oxidative stress in the CNS is a most exciting research field, emphasizing the importance of this mitochondrial protein in preserving the physiological homeostasis of the brain. Previous studies have shown that TSPO modulates the activity of mitochondrial ATP synthase, ROS generation, and the mitochondrial membrane potential (ΔΨm), indicative of regulation of mitochondrial pore opening, including voltage dependent anion channel (VDAC). This also entails mitochondrial Ca^2+^ and ATP release [8,12]. These are prerequisites considered part of the activation of the mitochondria-to-cell nucleus pathway for cell nuclear gene expression regulation [13]. Furthermore, the potentially accompanying cardiolipin oxidation in the inner mitochondrial membrane, simultaneous with Bax/Bak channel opening in the OMM, allowing for cytochrome c release, is part of the mitochondrial apoptosis cascade as one component for the induction of programmed cell death [13,14,15] (Figure 1). Both the modulation of cell nuclear gene expression and programmed cell death are part of progressing brain damage due to injury and disease [8,12,13,14,15].

## 3. 18-kDa Translocator Protein (TSPO) as a Venue for Diagnostic and Therapeutic Tools for Brain Disorders

It has long been recognized that TSPO expression is associated with neurological disorders [1,16]. It has been widely reported that overexpression of TSPO can result in microglial activation and is thereby involved in neurodegenerative disorders, such as Parkinson disease (PD), Alzheimer disease (AD), and Huntington disease (HD) [16,17,18]. Moreover, its association with the voltage-dependent anion channel (VDAC1) participates in initiating mitochondrial apoptosis, which includes the regulatory role of TSPO in ROS production and its downstream effects (Figure 1) [12]. TSPO’s contribution to these types of disorders was further revealed by discoveries about the increased binding potential (BP) to its ligands during induced neurodegenerative and neurocognitive disorders in animal models [19].

This evidence promoted the development of assays for possible changes in TSPO expression via its ligand BP in various CNS diseases and disorders by using positron emission tomography (PET) [20]. The synthesis of suitable radioactive-labeled TSPO agonists and tracers enabled accurate and non-invasive evaluation of TSPO as a biomarker for neuroinflammation. However, it should be stated that the first generation of radiotracers exhibited several limitations, such as the half-life of the radioactive elements and/or a low signal-to-noise ratio. This encouraged the production of more advanced TSPO PET ligands [21]. Thus, improvement of the effectiveness of TSPO ligands is crucial for broadening our knowledge of TSPO’s role in neuronal damage related to brain diseases and injuries.

The various cell types of the brain (in particular, neurons, astrocytes, and microglia) obviously play essential roles in the wide diversity of manifestations of brain disorders and diseases, and whether and how such brain disorders and diseases can be approached by TSPO ligands for diagnosis and treatments. In this Section 3, we describe some of the aspects of TSPO-related mechanisms in several brain cell types, i.e., neurons, astrocytes, and microglia, in relation to brain diseases and disorders.

### 3.1. Neurons and TSPO

In addition to TSPO being recognized as a marker for microglia and astrocyte activation, evidence exists about its expression in neurons in the CNS and peripheral nervous system (PNS) in responses to challenges. Thus, one can assume that variations in TSPO BP might not always be just associated with the microglia response in relation with pathophysiological processes in the brain but may also involve neurons more directly [22,23]. Early studies using immunocytochemical and imaging techniques demonstrated TSPO’s presence in immature neurons, as well as in regenerating neural tissues after a sciatic nerve injury. This correlated with TSPO regulation of programmed cell death, including the sensibility of developing/damaged neurites to pro-apoptotic signals [24,25]. Other studies, however, did not provide additional information regarding changes in neuronal TSPO expression in response to injuries [22,26,27]. Nonetheless, our own in vitro studies also suggest that neuronal TSPO may play a role in neuronal development, including axon generation [28]. Thus, further studies may reveal additional information regarding TSPO function in neurons subject to injury and disease. Of course, a question is whether such TSPO effects are due to TSPO expression in neurons in response to such brain diseases and injuries or are more dependent on TSPO in microglia and astrocytes (as outlined in Figure 2), or even other cells, such as oligodendrocytes and endothelial cells of the blood brain barrier (BBB).

### 3.2. Astrocytes and TSPO

While astrocytes have numerous functions mostly associated with preserving homeostasis in the CNS and also with normal neuronal development, for example, by their involvement in synapse formation and propagation of action potentials, it was also uncovered that they actively contribute to neuroinflammatory responses (reviewed by [29]) (see also Figure 2). Namely, research focused on astrocyte reactivity in neuronal disorders, which primarily consisted of immunostaining assays for the expression of glial fibrillary acidic protein (GFAP), an ubiquitously used marker in mammalian models, mounted enough evidence for the exploitation of astrocytes as potential biomarkers for traumatic brain injury (TBI), brain tumors, and stroke, as well as a wide range of neurodegenerative diseases [30,31,32]. The most recent investigations have revealed the existence of two phenotypes of astrocytes, termed A1 (inflammatory) and A2 (ischemic). The A1 phenotype is stimulated by proinflammatory mediators, such as tumor necrosis factor (TNF-α), interleukin beta (IL-β), and ROS production, and is abundantly present in most major neurodegenerative disorders, in contrast to the ischemia-induced phenotype (A2), which is upregulated by neurotrophic factors and primarily influences neuronal regeneration and reparation [29]. Single-cell genomics should help provide definitive answers as to the timing and possible simultaneous coexistence of multiple types of reactive astrocytes [29].

These findings present themselves as “windows of opportunity” for evaluating TSPO expression and BP in various brain disorders and diseases. One can assume this because of the similarities that they possess regarding the activation of microglial cells. Reports about increased binding of 3H-PK11195, 18F-DPA-714, and 11C-SSR180575 in astrocytes in vivo also give encouragement regarding the diagnostic potential of TSPO for neuroinflammatory diseases [33,34], even despite the lack of a significant correlation between the binding of this TSPO PET tracer and 11C-Pittsburgh compound B (11C-PIB) to GFAP-positive astrocytes, as found in post-mortem tissue from patients with AD [35,36]. Because of the low signal-to-noise ratio of 11C-PK11195 (full name: 1-(2-chlorophenyl)-*N*-methyl-*N*-(1-methylpropyl)-3-isoquinoline carboxamide) in in vivo studies, the use of this PET tracer for exploring TSPO BP in astrocytes is limited [37]. Nonetheless, a significantly increased presence of TSPO in astrocytes was determined in AD, in amyotrophic lateral sclerosis (ALS), and even in mild cognitive impairment (MCI), further supported by enhanced ligand binding of astrocyte-specific tracers, such as 11C-PIB and 11C-DED (full name: deuterium-l-deprenyl) (reviewed by [38]). These established findings open a window of opportunity for further investigations of TSPO as a biomarker for astrocyte activation occurring during neurodegenerative diseases.

In light of these assumptions regarding astrocytes, a recent report by Pannel et al. about the significantly higher TSPO expression in proinflammatory polarized cultured murine astrocytes measured by immunohistochemistry assays, together with their increased uptake of 18F-DPA-713 PET tracer, notably enhances the potential utility of TSPO imaging in neuroinflammation [39]. This study also revealed that astrocytes stimulated by the anti-inflammatory IL-4 do not share these TSPO-overexpressing features; however, the authors noted the possible differences in the phenotype of these cells to A2 astrocytes as reported by Liddelow et al. [29,39]. In summary, astrocytes can not only provide additional answers to the cellular mechanisms involved in the inflammatory response in the brain but are also worthy targets for TSPO PET ligand research.

### 3.3. Microglia and TSPO

Microglia cells present neuroinflammatory features by their abilities of transformation as induced by a pathologic event, such as acquiring an amoeboid morphology, which enables them to execute their response to harmful stimuli, including migration, proliferation, and phagocytosis [40] (Figure 2). Given their immediate activation even to the slightest change in the brain’s homeostasis, microglial cells are rightfully known as “sensors of brain integrity” [41]. Investigations have shown that there are two general phenotypes of activated microglia: The proinflammatory M1, whose classical neuroinflammatory response is mostly accomplished via the expression of major histocompatibility complex class II (MHC II) and the release of injury-mediating cytokines (TNFα, IL-6, IL-12), in contrast to the alternatively activated anti-inflammatory M2 microglia, which possess neuroprotective features achieved mostly by the secretion of cytokines (IL-4, IL-13) and growth factors, such as brain-derived neurotrophic factor (BDNF) as well as insulin growth factor (IGF) [42] (Figure 3). Based on this adaptive nature of microglia, and indubitably influenced by the local microenvironment, it has been discovered that besides participating in neuroinflammatory responses, microglia support many important physiological processes, such as brain wiring and maturation, including the pruning of excessive synapses and/or influencing neurogenesis [43,44,45]. The ubiquitous presence of active microglia offers the possibility of tracing various molecules and receptors present in their plasma membranes or subcellular compartments, further intensifying the search for suitable markers for different neuronal pathologies, characterized by activated microglial cells.

In view of these observations, much research has been carried out in order to track TSPO expression and ligand binding not just for uncovering the secrets of neuroinflammation but also in the hope of offering non-invasive techniques for locating them, thus providing a significant impact on the diagnosis of CNS diseases. From all the available in vitro studies up to this date, TSPO upregulation is indisputably confirmed in the M1 microglial phenotype [27,39] while enhanced TSPO expression in M2 has been less unequivocal [46]. Furthermore, it was also established that TSPO ligands 2-Cl-MGV-1 (full name: (2-(2-chlorophenyl) quinazolin-4-yl dimethylcarbamate)) and MGV-1 (full name: 2-phenylquinazolin-4-yl dimethylcarbamate) expressed anti-inflammatory effects by reducing IL-1β, IL-6, tumor necrosis factor alpha (TNF-α), and interferon gamma (IFNγ) levels in lipopolysaccharide (LPS) activated microglia [47]. Considering in vivo studies, TSPO expression mainly assayed by PET tracers’ uptake showed differences in the expression patterns in animal versus human microglia [48]. In addition, numerous studies have reported enhanced TSPO PET signals in various brain regions in subjects with different neurodegenerative diseases, such as PD, AD, and HD; as well as neurodevelopmental disorders, such as schizophrenia and autism; psychiatric disorders, such as major depressive disorder; and brain injuries (for example, reviewed in [49,50,51]).

Alterations of TSPO BP, as a result of injury and disease, and the influence of therapy in different neuronal pathologies in this context, are reviewed further in this paper.

## 4. TSPO in Neurodegenerative Diseases

As indicated above, it appears that TSPO presents enhanced expression in various cell types in brain areas affected by brain disease and injury. In this section, we discuss aspects of TSPO in neurodegenerative diseases (PD, AD, and HD) and psychiatric disorders, such as schizophrenia and autism spectrum disorder (ASD), as TSPO may present venues for diagnosis and therapy.

### 4.1. TSPO and Parkinson Disease (PD)

The early discoveries of α-synuclein (α-Syn) misfolding and oligomerization as a trigger for neurodegenerative diseases (such as PD and AD), which can provoke severe behavioral and cognitive impairment, encouraged investigations regarding the mechanisms related to the influence of α-Syn in the development of these diseases [53]. Essentially, the mechanisms responsible for neurodegeneration typically involve disruption of the molecular pathways associated with cellular degradation of α-Syn [54]. Its additional correlation with brain inflammation intensified research regarding elements of microglia activation, including interleukins, complement system members, and proinflammatory enzymatic pathways, that all are primarily associated with the promotion of microglia activation in affected CNS areas [55]. Efforts have also been targeted toward discovering the molecular regulatory mechanisms responsible for activating the cellular immune response, in particular via the evaluation of microRNA (miR) expression. For example, the roles of miR-155 and mir-124 in regulating inducible nitric oxide synthase (iNOS) activity and the expression of major histocompatibility complex class II protein (MHCII) and other inflammatory cytokines have been established [56,57].

Because these events are also closely associated with TSPO activity (Figure 3), this also suggests TSPO involvement in the neurodegenerative pathology of PD, potentially providing a non-invasive means for diagnosis. This approach is supported by the discovery of enhanced microglial activation obtained by PET binding of the TSPO ligand 11C-PK11195 in PD patients [58,59,60]. Additionally, increased TSPO expression was found in the midbrain of patients with rapid eye movements (REM), which can occur at early stages of PD, including accompanying sleep behavioral disorders, so called REM behavior disorders (RBD) [61]. However, research using second-generation PET radioligands, which also target genetic polymorphism influences (mostly single nucleotide polymorphism (SNP) rs6971) for TSPO binding affinity, gave less clear-cut results compared to those obtained with 11C-PK11195. For example, a study by Ghadery et al. using 18F–FEPPA showed that the significant differences in TSPO binding were associated with genetic predispositions presenting three groups: Low, mixed, and high-affinity binders (LABs, MABs, and HABs) [62]. However, within these three groups in the brain regions of interest, no significant alterations were discovered in the volume of distribution (VT) between healthy persons and PD patients, suggesting that genetic variations concerning TSPO binding affinity have little effect on PD risk [62]. Basically, identical observations were made by Varnas et al., who conducted similar research using 11C-PBR28 [63]. However, this study also noted that the significantly lower affinity of this ligand for LAB than 11C-PK11195, could be the reason for the decreased PET signal of 11C-PBR28 in PD patients [63]. Furthermore, it is known that second-generation TSPO tracers show additional limitations, such as a low quality of the images obtained for LABs, which limits their clinical value. Another disadvantage of these tracers is their interaction with lipids and proteins mostly derived from endothelial cells of the BBB, thereby limiting the concentration of ligand available for TSPO binding in the CNS, which ultimately results in “masking” of TSPO’s presence in certain brain areas [49,64]. Thus, care should be taking in interpretation of first- and second-generation PET ligands for TSPO labeling. 

Previously mentioned evidence about the protective activity of TSPO ligands in CNS disorders has initiated investigations about their effects concerning neurodegeneration. On this account, a study by Gong et al. showed that Emapunil, a novel synthetic TSPO ligand, significantly reduced dopaminergic loss in mice with induced PD by inhibiting the unfolded protein response and subsequently reducing cellular stress and apoptosis [65]. Additionally, the authors concluded that the neuroprotective effect of Emapunil might be associated with the partial alteration of neurosteroid levels, which can counteract neuroinflammatory responses [65]. Together with other studies applying TSPO ligands for the treatment of brain damage, this investigation showed promising results about exploiting TSPO as a possible therapeutic target for α-synucleinopathies [28,66]. However, greater efforts must be made to fully disclose the influence of TSPO ligands on dopaminergic degeneration.

### 4.2. TSPO and Alzheimer’s Disease (AD)

AD is a serious neurodegenerative disease characterized by memory loss and cognitive decline, predominantly associated with β-amyloid (Aβ) accumulation [67]. Recent studies suggest the involvement of transmembrane protein 21 KD (TMP21), also known as transmembrane emp-24 domain-containing protein 10 (TMED10). Given its important role in protein transport and preserving neuronal functions through the modulation of γ-secretase activity (an amyloid precursor protein (APP)-cleaving enzyme), dysregulation of TMP21 is considered one of the causes of AD development [68]. Namely, it was established that this protein is included in the regulation of molecular pathways associated with APP stability and Aβ aggregation mostly through inhibition of phosphoinositide-3-kinase [69]. It has also been suggested that TMP21 contributes to the hyperphosphorylation of Tau by upregulating the mammalian target of rapamycin (mTOR) pathway [70]. Moreover, as PD, AD is characterized by severe neuroinflammation derived from microglia and astrocyte activation, which is a plausible starting point for evaluating the possible involvement of TSPO expression [71]. Despite the satisfactory results about TSPO upregulation in different parts of the brain using autoradiography and PET imaging presented in the initial reports [72,73], later investigations using second-generation TSPO tracers showed discrepancies regarding TSPO binding [74,75]. Interestingly, similar observations were reported in animal models. Specifically, upregulation of TSPO binding was only observed in early stages of amyloidosis in the AD model of older mice [76]. Initial reports about increased binding of 3H-PK11195 in astrocytes also favor TSPO’s diagnostic potential for neuroinflammatory diseases, despite the lack of a significant correlation between the binding of this TSPO PET tracer and 11C-Pittsburgh compound B (11C-PIB) to GFAP-positive astrocytes presented in some studies [35]. However, the low signal-to-noise ratio of PK11195 was probably the main reason for these results. Thus, investigations about TSPO involvement in the activation of astrocytes with more specific TSPO ligands should offer different perspectives as to the location and timing of its overexpression in neurodegenerative disorders, such as AD. On the other hand, recent reports about TSPO ligand-mediated stimulation of pregnenolone synthesis in neuroblastoma cells overexpressing Aβ, thereby averting the effects ATP consumption and ROS production, i.e., TSPO-modulated mechanisms (Figure 3), highlight more possibilities for research about TSPO involvement in AD [77]. Therefore, future investigations should be directed towards revealing the oscillations in TSPO expression in astrocytes as well as treatment-related changes in the BP of TSPO in order to unveil its worth as a biomarker for neurodegeneration in AD. One aspect regarding possible therapeutic actions of TSPO ligands on neuroinflammation is to “check” its influence on neurosteroid synthesis.

### 4.3. TSPO and Huntington Disease (HD)

A third neurodegenerative disease to be mentioned here, HD, is an inherited and incurable neurodegenerative disease caused by trinucleotide repeat (CG) in the huntingtin (HTT)-encoding gene, which causes significant motor, cognitive, and psychiatric deficits [78]. This condition is also manifested by severe inflammation in the brain activated by the overexpression and aggregation of the mutated form of HTT [79]. The molecular mechanisms responsible for HD development include transcriptional dysregulation and impaired protein degradation [80]. Evidence about the mitochondrial dysfunction associated with AD pathology poses the question of possible TSPO involvement in HD as well, like in AD and PD. Studies involving in vivo microglial PET imaging suggested significantly increased uptake and BP of 11C-PK11195 in patients with HD versus HC (healthy controls) in all of the analyzed brain regions [81,82]. In contrast with the results of TSPO BP in AD and PD, investigations with second-generation radiotracers, such as 18F-PBR06 and 11C-PBR28, confirmed previous findings, thereby acknowledging TSPO as a potential marker not only for diagnosing HD but also for evaluation of the effect of treatment [83,84]. Vainshtein et al. and Veenman et al. showed an ameliorating effect of the TSPO ligand 2-Cl-MGV-1, which was vastly superior to PK 11195 and MGV-1 in a transgenic mouse model for HD [28,85].

### 4.4. TSPO in Psychiatric Disorders

Psychiatric disorders, such as schizophrenia and autism spectrum disorder (ASD), show emotional and cognitive dysfunction, as neurodegenerative diseases, such as PD, AD, and HD, also can do. Nonetheless, as the name suggests, neurodegeneration is not the major hallmark of psychiatric disorders. However, microglial activation may occur with psychiatric disorders. Thus, it appears worthwhile to investigate TSPO’s involvement in psychiatric disorders.

#### 4.4.1. TSPO in Schizophrenia

Psychiatric disorders, such as schizophrenia, major depression disorder (MDD), and bipolar affective disorder (BPAD), are often identified as mental declining syndromes typically associated with significant disturbances in cognition and emotion. These diseases that are mainly characterized by impaired neural development and defective cellular signaling manifest themselves by thought disorders, including delusions and hallucinations, and also by fasciculation [86]. Ever since the discovery of the severity of these mental disorders, constant efforts have been made to establish effective treatment, including massive exploration to uncover the molecular pathways involved in their underlying pathology. For example, researchers have targeted the balanced translocation t(1;11) (q42;q14.3) in the disrupted-in-schizophrenia 1 (DISC1) gene, which induces synthesis of modified DISC1 protein, a primary risk factor for neuropsychiatric pathology [87]. A study by Shao et al. showed that expression of a truncated form of DISC1 manifests itself by reduced reversal learning experience, increased glutamatergic transmission, and abnormal development of neuromuscular junction in *Drosophila melanogaster* [88]. In parallel, investigations have also revealed that the normal expression of DISC1 is influenced by several other proteins, including dysbindin, pituitary adenylate cyclase-activating polypeptide (PACAP), DISC1-binding zinc finger protein (DBZ), fasciculation and elongation protein zeta-1 (FEZ1), stathmin1, and kendrin [89,90,91,92]. This suggests their possible contribution to the pathology of psychiatric diseases. Indeed, following their discoveries, reports showed that the upregulation of DISC1–FEZ and DISC1–kendrin interactions stimulates neural development. In addition, PACAP is mainly responsible for controlling neurite growth by dissociating DISC1–DBZ interactions and inhibiting stathmin 1 overexpression [92]. Additionally, TSPO is known to affect neuronal development and neurite growth [28,85] (see Figure 2). Kubota et al. connected dysbindin loss with schizophrenia, indirectly associating its normal expression with regular neurogenesis [90]. Furthermore, DISC1 and DBZ have opposite effects in the process of oligodendrocyte differentiation [89,91].

Regarding the rates of inflammation, several publications have reported increased microglial activation in post-mortem patients with schizophrenia [93,94], accompanied by elevated levels of cytokines in circulation [95], which may suggest the involvement of TSPO (Figure 3). However, it should be taken into account that changes in peripheral markers may not necessarily be the result of microglial activation. Thus, further investigations are needed to more conclusively connect neuroinflammation with this type of disorder [96]. We believe that possible alterations of TSPO expression may provide additional evidence concerning the inflammatory rates in these diseases. Unfortunately, studies have revealed diverging effects regarding TSPO expression in patients with schizophrenia. Explicitly, while early reports suggested an increase in BP in patients suffering from this disease [97], the use of more recent PET tracers, such as 18F-FEPPA or 11C-DAA1106, gave no significant changes regarding TSPO binding [98,99]. However, these contradictions could be explained by factors, such as disease duration and medication use, which are possibly associated with the varying severity of the immune response in different stages of the disease [100]. Notably, it should be considered that antipsychotics and mood-stabilizing medications also tend to decrease glial activation [101]. In accordance with these discoveries, Setiawan et al. showed increased TSPO density measured by the volume of distribution (VT) in medication-free mood disorder patients with moderate to severe depressive symptoms. Furthermore, a connection was established between this parameter and the severity of depression quantified via the Hamilton Depression Rating Scale (HDRS) [102]. However, a positive correlation was not reported between serum inflammatory markers and TSPO density measured in the prefrontal cortex, anterior cingulate cortex, and insula [102]. On the other hand, Collste et al. reported significantly decreased levels of TSPO BP in various brain regions in drug-naive first-episode psychosis patients using the 11C-PBR28 radioligand [103]. This is probably due to an impaired neuroinflammatory response or late immune activation [103]. Consequently, the final word regarding neuroinflammation and TSPO in relation to schizophrenia has not been said yet.

In light of these discoveries, the study of Pouget et al. about the possible correlation between TSPO and schizophrenia showed no association between the disease and any of the known single nucleotide polymorphisms (SNPs) of this gene in large cohorts of diagnosed patients receiving various antipsychotics. Furthermore, it was established that polymorphisms of TSPO were not directly accountable for a successful response to treatment according to the Brief Psychiatric Rating Scale (BPRS) score. Interestingly, these authors reported a significant connection between rs6971 SNP and weight gain in patients treated with atypical antipsychotics, such as clozapine, olanzapine, and risperidone, confirming the findings of the metabolic effects of these medications [104,105]. Possibly, this association could be the result of previously reported involvements of TSPO in glucose homeostasis, as suggested by the lowering of the glucose concentration by the TSPO ligand 11C-PK-11195 [106]. Therefore, a mutation in the TSPO gene might disrupt this “glycolytic” effect and therefore be responsible for the significant weight increase.

#### 4.4.2. TSPO in Autism Spectrum Disorder (ASD)

ASD entails a group of grievous neurodevelopmental conditions typically manifested by persistent deviations in social interactions and communication, accompanied by unusual behavioral disorders. Thus, subjects suffering from ASD present impaired cognitive functions, typically displayed by compromised executive functions, reduced perceptual abilities, and poor mental skills [107]. The etiologies of these disorders are mainly genetic, but the latest research indicates that environmental factors are also involved in the initiation or aggravation of the symptomatology of ASD. This may underly the drastic enhancement of the prevalence of ASD, even up to an epidemic level, especially among young adults [108]. Despite the enigma of the exact pathogenesis of ASD, numerous studies have indicated that neuroinflammation might be one of the key factors for its development. The latter is assumed because activated microglia take part in the elimination of redundant synapses, a process known as synaptic pruning [109,110]. Thus, we believe that TSPO as a regulator of neuroinflammation and neuronal development may be involved in ASP. These changes in brain “physiognomy” with ASD are associated by alternating the inhibitory role of the mTOR pathway on neuronal autophagy by mutations in the genes involved in its regulation [111]. In line with these investigations, it is proposed that the induction of the M2 microglial phenotype plays a potential part in modulating the process of synaptic pruning by stimulation of mTOR. In this context, the M2 microglial phenotype, which possess neuroprotective roles through the secretion of cytokines and growth factors, stimulates mTOR upregulation [112]. The presence of both microglial phenotypes (the proinflammatory and M1 and anti-inflammatory M2) has been confirmed by several post-mortem studies of ASD patients [113,114]. In accordance with these results, it can be acknowledged that a possible increase in IL-4 and IL-13 secreted from T helper 2 (Th2) lymphocytes or mast cells formed as a result of an allergic reaction, eczema, or asthma could result in M2 microglial polarization and might contribute to ASD symptomatology, due to the dysregulation of neurodevelopment.

Additional evidence of microglial activation in ASD was provided by modulations of TSPO expression using [3H]PK-11195 as a PET tracer. Namely, Suzuki et al. conducted a voxel-based analysis, which showed that patients suffering from ASD had higher radioligand uptake than the corresponding healthy controls in a wide range of brain areas, including the cerebellum, brainstem, and the frontal, temporal, and parietal regions. However, no significant correlation was found between TSPO BP and the clinical features of the disease that were determined via standard diagnostic tests for autism [115]. In the absence of conclusive diagnostic tools for autistic syndromes, TSPO expression could be included in their detection. Additional research should be carried out to establish the possible variations of TSPO BP in ASD patients with and without therapy.

### 4.5. TSPO, Neurodegeneration, and Psychiatric Disorders in Short

From the descriptions and discussions above, it appears that in association with inflammatory responses, including microglial activation, enhanced TSPO levels accompany neuropathologies that occur in the brains of people suffering from neurodegeneration and psychiatric disorders. The same is true for animal models of these conditions. The knowledge that TSPO is involved in the modulation of several general functions that are part of these disorders and diseases, such as programmed cell death, gene expression modulation, microglial activation, and neuronal (re)generation, suggests that TSPO plays essential roles, and can be targeted for treatment and diagnosis. For an illustration of these TSPO-associated functions, see Figure 1, Figure 2, Figure 3 and Figure 4.

## 5. TSPO and Alcohol Use Disorder (AUD)

Alcohol consumption contributes to the development of psychiatric, neurological, social, and physical diseases and disorders (just a few examples include depression; suicidality; liver cirrhosis; health-risk behaviors, including driving accidents; and other accidents due to risk taking and poor judgement after drinking) [116]. Some of these effects are described in more detail in this section. Alcohol use disorders (AUDs) are the third cause of preventable death in the United States [117]. Indeed, alcohol consumption is a serious global healthcare problem with severe consequences for the human population [118]. In simple terms, alcohol overdose can be considered neurotoxic, i.e., causing brain damage that can lead to psychiatric disorders as well as neurodegenerative disease. Problems caused by alcohol consumption include the disruption of normal brain activity, and dependency on the dose, duration, and pattern of exposure, manifests with cognitive deficiencies, compromised decision-making, and impaired motor skills [119]. Moreover, alcohol intoxication and/or withdrawal can result in debilitating morbidities, such as anxiety and depression [120]. Concerning these grave implications together with the increasing trend of alcohol abuse over the recent years, significant efforts have been made to investigate the pathophysiological and molecular pathways involved in the response of the organism to alcohol consumption [121,122]. Even more disturbing, it seems that the adolescent population is more vulnerable to chronic alcohol exposure than adults. Because alcohol directly inhibits excitatory synapses in the brain, it can produce depressant effects in critical brain regions that control responses, such as behavior and judgment, movement and coordination, sleep, and vital functions. In particular, major regions distributed throughout the brain that can be negatively impacted by alcohol include the ventral striatum and prefrontal cortex, the cerebellum, the reticular-activating system, and the medulla (extensively reviewed in [123]).

Ethanol is a small molecule that can easily cross the cell plasma membrane and thus diffuse rapidly throughout the CNS, affecting brain tissues, depressing neuronal activity, and interfering with neuronal communication [123]. For example, toxic metabolites of ethanol, mostly acetaldehyde, disrupt the protein structure, producing protein adducts, which inevitably trigger cellular injury in brain regions, such as those mentioned above [124,125]. On the other hand, these protein adducts could also be the result of the significantly enhanced concentration of lipid peroxidation products derived from an ROS imbalance in the cells in question [126,127]. This presence of protein adducts in the brain is also the cause for the production of inflammatory cytokines, which greatly exacerbates the injury caused by ethanol [128]. Given this “oxidative nature” of neuronal damage, it can be assumed that the use of antioxidants would be able to ameliorate and/or reverse some of the negative effects caused by ROS overproduction. These characteristics are also known to be part of TSPO-driven functions, e.g., see Figure 1.

Within neural cells, several structures are modified by ethanol toxicity; among these, the mitochondrion is an organelle that is particularly sensitive. Although the exact mechanisms by which alcohol exerts its toxicity in the brain are not fully clear, it is widely accepted that it includes oxidative stress and promotes neuroinflammation. Mitochondria are the main producer of ROS in the brain and contribute to the inflammatory processes. As described above, it is well known that TSPO is an important mitochondrial protein, involved in ROS generation, microglial activation, and gene expression, that is associated with normal physiology as well as disease and injury states [123]. Thus, we work from the premise that the mitochondria, including TSPO, act as an important mediator in alcohol-induced toxicity, positioning it as a potential therapeutic target for the treatment of diseases associated with AUD during adolescence and their persistence into adulthood.

The mitochondrion is an organelle whose main function is to carry out aerobic cellular respiration, producing energy in the form of ATP [129]. Neurons possess a large number of mitochondria throughout the dendrites and axons, including the synapse [130]. Interestingly, in this context, the human brain consumes approximately 20% of the total energy pool produced by mitochondria in the body [131]. In addition, mitochondria are also important regulators of ROS production, redox cell balance, calcium homeostasis, and cell death [4,132,133], and other functions described further below. Indeed, alcohol can affect neurons by altering the mitochondrial function via impairment of the electron transport chain, generating oxidative stress and reducing energy production. Neurons continually form mature synapses, and mitochondrial alcohol toxicity can inhibit this process and could even induce neuronal apoptosis [134].

An increase in mitochondrial membrane permeability is a characteristic symptom of mitochondrial impairment in the brain [135]. A well-known element for this mechanism is the mitochondrial permeability transition pore (mPTP), which is a nonspecific high-conductance channel that is permeable to molecules up to 1.5 KDa [136]. The mPTP is considered to be formed by, or part of, a multiprotein complex. Just four proteins are mentioned here for illustration: (1) The voltage-dependent anion channel (VDAC) is located in the OMM. As its name implies, VDAC is a high-conductance channel, thus it is essentially the core of the mPTP. (2) The adenine nucleotide transporter (ANT), located on the inner mitochondrial membrane, which, similar to VDAC, can also work as a channel for small molecules. (3) Cyclophilin D (CypD) a mitochondrial matrix protein. Lastly, (4) very much to our interest, the translocator protein TSPO, an OMM protein, is very closely associated with VDAC and ANT [1,2,137]. mPTP is opened by mitochondrial calcium overload, oxidative stress, changes in the pH, and ROS [11,138]. Transient openings of this pore induce the release of excess calcium accumulated in the mitochondrial matrix [135]. However, when mPTP is constantly open it causes collapse of the mitochondrial membrane potential (ΔΨm), leading to opening of the Bak/Bax channel and thus allowing for the release of pro-apoptotic factors, leading to cellular death [14,15,139]. Our studies have shown that TSPO is a major component for the induction of programmed cell death and modulation of gene expression via opening of the mPTP (e.g., see Figure 1).

In this context, Wistar rats exposed for 1 h to 1 mL of absolute ethanol showed increased oxidative stress markers in the brain [140]. Interestingly, neuronal cultures present permanent mPTP opening after treatment with ethanol, leading to neuronal apoptotic death [132,141]. In astrocytes, ethanol induces a transient opening of the mPTP [141]. We assume that such a transient opening of the mPTP (or long-term slight depolarization of the ΔΨm), as opposed to permanent collapse of the ΔΨm that can be observed in other cell types, may lead to the modulation of cell nuclear gene expression [13]. Altogether, these changes in the ΔΨm and increased ROS production can result in decreased ATP production [142,143].

As detailed below, the mitochondrial protein TSPO, putatively part of the mPTP, is closely involved in the various known mechanisms that are part of brain damage caused by ethanol. For example, we know that excitatory synapses by their activity modulate TSPO function, and vice versa, the latter in particular via gene expression modulation [8]. Importantly, TSPO modulates processes, such as ROS generatio; iNOS activation; Ca^2+^ homeostasis; ATP synthesis; heme production; programmed cell death; cell cycle; neuronal regeneration, including axonal regeneration, neuroinflammation, and angiogenesis; and also modulates gene expression related to these processes [9,10,28,66,85,144,145,146,147] (Figure 1, Figure 2, Figure 3 and Figure 4). A compilation of some of the neuroprotective effects of TSPO ligands and the related molecular biological mechanisms of TSPO are presented by various papers in two Special Issues, respectively in the International Journal of Molecular Science (TSPO and Brain Disorders) and in the journal Cell (Differential Regulation of Glial and Neuronal Functions by TSPO). This short list of publications regarding TSPO functions shows that the majority of the processes related to alcohol-induced brain damage referred to below can be influenced by TSPO and its ligands.

Alcohol intoxication can lead to neurodegenerative processes, which entail oxidative signaling and excitotoxicity, and can lead to motor and behavioral problems [148,149,150]. Activation of inflammatory pathways in the brain is also considered part of ethanol toxicity [151,152,153]. Thus, such alcohol consumption can produce major problems in the brain, for example, damage to brain cells, even cell death, and neuroinflammatory responses, thus presenting a main risk for brain pathologies and disruptive behaviors.

Several pathways leading to the damage of brain cells due to alcohol in general and binge drinking in particular are known. As described above, in general, these pathways appear to be associated with mitochondria, including the involvement of TSPO. Acetylaldehyde is part of ethanol metabolism, and promotes ROS formation and induces apoptotic cell death. Chronic ethanol treatment in rats promotes the production of ROS and reactive nitrogen species (RNS). These events favor lipidic peroxidation of neural membranes, increasing the activity of NADPH oxidase and NOS enzymes and reducing the activity of antioxidant enzymes, including superoxide dismutase (SOD), catalase (CAT), glutathione peroxidase (GPX), and glutathione reductase (GR). As discussed above, TSPO is typically involved in such processes [123] (see Figure 1).

Another mitochondrial function is the maintenance of calcium homeostasis. Finally, all of these events result in changes in the expression of mitochondrial complexes I and V (ATP synthase), and a bioenergetics deficit as is indicated by low ATP concentrations. Altogether, these negative mitochondrial effects, mediated by ethanol exposure, include cytochrome c release from the mitochondria. As mentioned above, mitochondrial cytochrome c release can be regulated by TSPO activation [123].

### AUD Excerpt: Mitochondrial Involvement, Including TSPO

Excerpted in short, ethanol exposure induces negative changes in mitochondrial function. These alterations include but are not restricted to: (1) Increased ROS production, (2) deregulation of calcium homeostasis, (3) mitochondrial respiration impairment, (4) reduced ATP production, and (5) mPTP opening. All these events, in turn, can result in neuronal death [123]. Thus, it appears from these various studies that mitochondria play a fundamental role in alcohol toxicity during binge drinking consumption, and most importantly, mitochondrial damage persists and can progress over time, even until adulthood when ethanol is absent.

As the associated functions and protein expression discussed above are also known to be regulated by the mitochondrial protein TSPO, one can readily assume that TSPO is involved in ethanol toxicity. Indeed, recent experimental studies in non-human primates, rats, and insects have demonstrated that TSPO is associated with the effects of ethanol. Thus, experimental studies in animals favor this assumption that TSPO is involved in the effects of AUD. Results regarding TSPO’s ligand-binding potential suggest higher uptake of TSPO tracers 11C-PK11195 and 11C-DAA1106 in rats after intrastriatal injection of alcohol using autoradiography assays [154]. In more detail, these studies showed increases of TSPO expression in astrocytes as well as microglia in rats after ethanol-induced brain injury. In another study, also in rats, ethanol-induced brain injury resulted in enhanced microglia activation, including enhanced TSPO expression [155]. Enhanced TSPO expression indicative of glial activation due to ethanol exposure could persist for at least several months in baboons [156]. However, the findings of lower 11C-PBR28 uptake in patients with alcohol use disorder compared to healthy controls in medium-affinity binders, and no differences between high-affinity binders suggest that microglia might not be involved in alcohol intoxication. Similar results were also noticed in the brains of ethanol-exposed rats vs. control rats [157,158]. In *Drosophila*, TSPO knockdown in neurons results in increased sensitivity to ethanol sedation [159]. Thus, the studies presented in Section 5 indicate that the mitochondrial TSPO present in various cell types in the CNS might provide an important target for treating AUDs.

## 6. TSPO and Traumatic Brain Injuries (TBIs)

Traumatic brain injuries (TBIs) are mostly characterized as disturbances in the normal physiology and structure of the brain caused predominantly by extrinsic mechanical insults. These conditions are often associated with neuronal, axonal, and vascular damage, which provoke microglial activation, including cytokine release and ROS generation, as primary self-defense responses of the brain [160]. Injury induces interlooping vicious cycles of neuronal and astrocytic cell death, cell damage, and cell activation driving microglia activation, which in turn causes additional progressive cell damage and cell death [29,161]. First of all, by its mechanical force alone, an impact can induce neuronal and astrocytic damage and, depending on the severity of the impact, may include astrocytic death, neuronal death, and axon degeneration. Subsequently, the death, damage, and activation of neurons and astrocytes typically induce microglial activation [162]. Acute microglial activation can serve to protect neurons and astrocytes from additional damage and/or enable the reversal of damage [163]. However, depending on the severity of the TBI, chronic microglial activity after TBI can induce progressive brain damage. Thus, microglial activation can serve to either protect brain tissue from damage, or to cause brain damage [29]. Further, the self-repairing mechanisms of neurons and effects originating from activated microglia and/or astrocytes can help neurons to survive and even regenerate axons, while the activation of mechanisms associated with programmed cell can cause brain tissue damage (Figure 2 and Figure 3) [29].

In addition to the wide variations in the severity of injury and subsequent disease, the complexity of the interaction of various brain cell types contributes to the well-known difficulties observed with the development of reliable treatments for TBI [164]. Nonetheless, knowledge regarding impaired Ca^2+^ homeostasis and decreased levels of ATP, leading to mitochondrial dysfunction in case of TBI, and the triggering of microglial activation (Figure 4) might offer valuable “clues” not only to the molecular mechanisms involved in the progression of TBI but also to its treatment [165]. Therefore, it is safe to assume that ligand-mediated TSPO tracking may be a valuable diagnostic tool for the severity and for localization of TBI, apart from its use as a therapeutic target, considering the effects of some radiotracers developed over the years. In line with these assumptions, it was reported that high-affinity ligands of TSPO, such as PK-11195, by themselves can reduce proinflammatory gene expression of cyclooxygenase-2 (COX-2), TNF-α, and IL-6 after lipopolysaccharide (LPS) stimulation [166], with a concomitant reduction in the number of activated microglia, associated with mitigated degeneration [167]. A recent study confirmed the therapeutic effects of two more novel TSPO ligands: 2-Cl-MGV-1 and MGV-1, which can alleviate the inflammatory response in microglial cells in culture via a reduction of COX2 and iNOS expression, as well as nitric oxide (NO) generation [66]. It is also known that 2-Cl-MGV-1, in addition to a reduction of microglial activation, prevents the cell death of astrocytes and neurons, and promotes neurodifferentiation in cell culture and in animal models through pathways involving the inhibition of cellular apoptosis and modulation of gene expression (as determined by genomics) [13,28,85]. Furthermore, radioligand-labeled research revealed the protective effects of moderate-affinity TSPO ligands, such as Etifoxine, against brain injury and brain disorders [168,169]. Thus, TSPO has become the target of research into high-affinity as well as low-affinity therapeutic ligands against various pathologies of the CNS, leading to a plethora of patents [14,65,170,171]. Interestingly, although counterintuitive, from the results of various studies, it can be assumed that low- and moderate-affinity TSPO ligands are at least as efficacious as high-affinity ligands, while showing far fewer risks of side effects [28,47,66,172,173].

Research concerning TSPO ligand uptake in animal models of TBI overwhelmingly supports the findings of its role as a microglia/macrophage activator. Specifically, studies have revealed that the increases of BP using 11C-PK11195 can be associated with both the time and type of brain injury [174,175,176]. Investigations using second-generation ligands fully support the results of the increased uptake in different types of trauma but are in contrast with the findings of increases of TSPO expression early after traumatic events [50]. Namely, studies have observed significant elevations of TSPO BP using 18F-DPA-714 as early as 2 days after the injury, with a proportional increase for at least 16 days in animal models [177,178]. The same results were obtained with 3H-PK11195 and 18F-fluoroethyl-DAA1106 in brain regions not primarily affected by cranial damage [179,180]. Findings that TSPO is upregulated in both M1 and M2 microglial phenotypes may not necessarily undermine its diagnostic value, given that plasma/serum markers for TBI, such as cytokines, GFAP, or BDNF, have a significantly shorter half-life and are measured using invasive techniques [181].

## 7. TSPO and Stroke

In a review, Schumacher et al. discussed the neuroprotective effects of progesterone in traumatic brain injury (TBI), stroke, spinal cord lesions, and motoneuron disease. Progesterone may promote neuroregeneration by several different actions: By reducing inflammation, swelling, and apoptosis; increasing the survival of neurons; and the formation of new myelin sheaths. We now know that these functions are also affected by TSPO, as discussed in this review, and the cited papers. Thus, the possible application of TSPO ligands may locally increase the synthesis of steroids by neurons and glial cells to provide neuroprotective and neuroregenerative effects in various CNS injuries, including stroke [182]. Microglial activation is an important component of the inflammatory response to ischemic stroke. Price et al. used 11C-PK11195 for PET analysis to identify significant binding in core infarction, contralateral hemisphere, and within a defined peri-infarct zone in brains of patients with ischemic stroke. They found minimal activation of microglia before the timepoint of 72 h after stroke. After that, the binding potential increased in the core infarction, peri-infarct zone, and contralateral hemisphere till 30 days. They concluded that this may represent a window of opportunity extending beyond the traditionally reserved time windows for therapies [183].

It is known that TSPO is present in arterial plaques, which are a characteristic of atherosclerosis that can be a contributing cause for ischemic stroke [5,6]. TSPO radiotracers, other than ^11^C-PK11195 for PET analysis, have been applied to stroke research. A TSPO molecular imaging biomarker 11C-vinpocetine showed different time-dependent decreases between microglial activation in the peri-infarct zone and the ischaemic core of stroke [184]. In addition, increased microglial activation in the peri-stroke region was seen for several weeks after the insult [184]. Various attempts were and are being made to develop TSPO radioactive tracers for adequate molecular imaging methods to assess the efficacy of stroke treatments. For example, overexpression of TSPO in the monocytic lineage as well as astrocytes that follows focal cerebral ischemia can be imaged with PET of the second-generation TSPO-selective radioligand 18F-DPA-714 [185]. In this way, the effect of minocycline, a broad-spectrum antibiotic, was tested on the inflammatory reaction induced in a rat model of stroke [185]. In vivo PET imaging showed a significant decrease in 18F-DPA-714 uptake at 7 days after cerebral ischemia in rats treated with minocycline with respect to saline-treated animals. Surprisingly, while apparently effective on the inflammatory response after stroke, minocycline treatment had no effect on the size of the infarcted area [185]. The novel TSPO tracer 18F-GE-180 was also tested in a rat model of stroke [186]. 18F-GE-180 uptake was 24% higher in the core of the ischemic lesion and 18% lower in the contralateral healthy tissue than that of 11C-PK11195 uptake (i.e., a 1.5 ± 0.2-fold higher signal to noise ratio) [186]. Following transient middle cerebral artery occlusion (MCAO) in rats, longitudinal in vivo magnetic resonance (MRI) and PET imaging studies were carried out with the novel TSPO radiotracer 2-18F-VUIIS1008 99 *N*-diethylacetamide (18F-VUIIS1008), and 18F-DPA-714 (full name: (*N*, *N*-diethyl-2-(2-[4-(2-fluoroethoxy)-phenyl]-5,7-dimethyl-pyrazolo[1,5-a]yrimidin-3-yl)-acetamide)) [187]. 18F-PA714 uptake showed a mild uptake increase compared to 18F-VUIIS1008 in TSPO-rich ischemic brain regions [187]. Combined with other studies, it appears that 18F-GE-180 is more effective than both 18F-DPA714 and 18F-VUIIS1008. Chaney et al. designed a study to directly compare two promising second-generation TSPO tracers, namely 11C-DPA-713 and 18F-GE-180, at acute and chronic time points after ischemic stroke induced by distal middle cerebral artery occlusion (dMCAO). Importantly, a significant correlation was identified between microglial/macrophage activation and the 11C-DPA-713-PET signal, which was not evident with 18F-GE-180. No significant correlations were observed between TSPO PET and activated astrocytes (GFAP immunolabeled) [188]. Thus, 11C-DPA-713 and 18F-GE-180 PET enable the detection of neuroinflammation at acute and chronic time points after cerebral ischemia in mice. Chaney et al. concluded further that 11C-DPA-713 PET reflects the extent of microglial activation in infarcted dMCAO mouse brain tissue more accurately than 18F-GE-180 [188]. In a study comparing second-generation TSPO tracers 18F-GE-180 and 18F-DPA-714 to 11C-PK11195 in a rodent model of subtle focal inflammation, 18F-GE-180 showed a higher core/contralateral ratio and binding potential when compared to 11C-PK11195 while 18F-DPA-714 did not [189].

Atherosclerosis is a common and serious vascular disease predisposing individuals to myocardial infarction and stroke [5,6]. Intravascular plaques, the pathologic lesions of atherosclerosis, are largely composed of cholesterol-laden luminal macrophage-rich infiltrates within a fibrous cap [190]. The ability to detect those macrophages non-invasively within the aorta, carotid artery, and other vessels would allow physicians to determine the plaque burden, aiding the management of patients with atherosclerosis. The 125I-iodo-DPA-713 (iodoDPA) TSPO radioligand selectively targets macrophages [191]. Single-photon emission computed tomography (SPECT) images showed the focal uptake of the radiotracer at the aortic root in all ApoE^−/−^ mice (Apolipoprotein E^−/−^ mice), while the age-matched controls were nearly devoid of radiotracer uptake [191]. Applying the second-generation radioligand 18F-GE180 TSPO, for PET, showed distinct patterns of upregulation of TSPO 18F-GE180 uptake in the carotid plaques of stroke patients, suggestive of activated macrophages’ infiltration [192]. It has also been suggested by Chen et al. that plasma TSPO may be intimately linked with disease progression and worse functional outcomes in acute ischemic stroke patients [193].

As the mitochondrial TSPO is obviously a protein that is involved in various aspects of stroke, it was recognized that it can be targeted for treatment of this condition. Li et al. investigated the effect of the TSPO ligand Etifoxine on neuroinflammation and brain injury after ischemia/reperfusion [194]. Etifoxine significantly attenuated neurodeficits and infarct volume after middle cerebral artery occlusion (MCAO) and reperfusion. The attenuation was pronounced in mice subjected to 30, 60, or 90 min of MCAO [194]. Etifoxine reduced the production of proinflammatory factors in the ischemic brain. In addition, Etifoxine treatment led to decreased expression of IL-1β, IL-6, TNF-α, and iNOS by microglia. Notably, the benefit of Etifoxine against brain infarction was ablated in mice depleted of microglia using a colony-stimulating factor 1 receptor inhibitor [194]. These findings indicate that the TSPO ligand Etifoxine reduces neuroinflammation and brain injury after ischemia/reperfusion [194]. This presents one reason why the therapeutic potential of targeting TSPO warrants further investigation in ischemic stroke.

In another model, after applying permanent cortical ischemia by distal middle cerebral artery occlusion (dMCAO) in rats, the effects of the TSPO ligand 2-Cl-MGV-1 were assayed on poststroke cognitive deficits, neuronal damage, mitochondrial apoptosis, and secondary damage in the non-ischemic, i.e., indirectly affected ipsilateral thalamus and hippocampus, after this cortical infarction [172]. This model induces stroke limited to the parietal neocortex, avoiding direct damage to the thalamus and hippocampus. In this permanent cortical ischemia model, infarct volumes did not significantly differ between the vehicle and 2-Cl-MGV-1 groups. There were more neurons and fewer glia in the ipsilateral but non-ischemic thalamus and hippocampus in the vehicle groups than in the sham-operated group 7 and 14 days post-dMCAO [172]. 2-Cl-MGV-1 significantly ameliorated spatial cognitive impairment and decreased neuronal death and glial activation when compared with the vehicle treatment. Mitochondrial transmembrane potential depolarization and cytoplasmic release of apoptosis-inducing factors and cytochrome c was prevented within the thalamus. Caspase cleavage and the numbers of terminal deoxynucleotidyl transferase dUTP nick-end labeling or Nissl atrophic cells were reduced within the thalamus and hippocampus [172]. This was accompanied by upregulation of B-cell lymphoma 2 (BCL-2) and downregulation of Bax [172]. Thus, 2-Cl-MGV-1 appears to reduce neuronal apoptosis via mitochondrial-dependent pathways and attenuates secondary damage in the nonischemic thalamus and hippocampus, potentially contributing to amelioration of cognitive deficits after cortical infarction [172].

Thus, TSPO based treatments of brain disease and brain injury, including stroke, may become increasingly more successful [9,169,172,182,195,196]. Further, as mentioned, non-TSPO targeting agents may also reduce the inflammatory response after stroke [182,185]. Thus, it is becoming more essential to better understand the recovery processes after stroke, and also of other brain injuries. To study this aspect for stroke, PET was combined with matrix-assisted laser desorption/ionization mass spectrometry (MALDI-MS) imaging [197]. Three months post-stroke, PET imaging with 18F-DPA-714 shows minimal detection of neurodegeneration and neuroinflammation, indicating that the brain has stabilized [197]. However, MALDI-MS images reveal distinct differences in lipid distributions (e.g., phosphatidylcholine and sphingomyelin) between the scar and the healthy brain, suggesting that recovery processes are still in play three months post-stroke [197].

## 8. Conclusions and Implications for Future Studies

Despite the various and sometimes conflicting findings of TSPO ligand uptake concerning different CNS disorders, it remains one of the most promising “tools” for their non-invasive diagnosis. Given that TSPO expression is not only altered merely as a result of microglial activation but could also be an indicator of disease severity and/or progression, TSPO has also been considered for the diagnosis of neurodevelopmental disorders. Furthermore, TSPO ligands as biomarkers may teach us how TSPO is involved in processes related to brain disorders, including its regulation of various mitochondrial functions. Moreover, the changes in TSPO expression observed with brain disease and brain injury suggests that TSPO may also be a target for treatments. This presents a plausible starting point for translational studies and future clinical trials, for example, including longitudinal studies involving the quantification of TSPO BP through new generations of PET tracers. This may uncover functional pathways involved in the various pathologies, thereby leading to the development of novel drugs interacting with such TSPO-associated functional pathways. Such pathways may include programmed cell death, inflammatory response, regeneration, etc., i.e., both damaging and repairing pathways. Furthermore, at cellular levels, TSPO will show different function characteristics in microglia, astrocytes, neurons, oligodendrocytes, endothelial cells, etc. Thus, the functional effects of TSPO activity in these various cell types should be differentiated. Finally, of course, the effects on tissue and on brain tissue as well behavioral and cognitive aspects should be studied. One last statement, as TSPO can be considered a protein serving homeostasis at cellular, tissue, and whole organism levels, one may hope that side effects may be avoided by applying the appropriate TSPO ligands. This appears to be so, because in animal models, TSPO ligands, such as PK11195, Etifoxine, Emapunil, and 2-Cl-MGV-1, appear to be successful agents for treating brain disease and injury, in particular the neurodegenerative and neuroinflammatory components. We hope that this review presents a sufficient overview of TSPO functions in relation to an organism’s varied responses to brain damage and brain disorders. We believe that the idea about TSPO as a potential target for the treatment of brain damage, as expressed by the various studies summarized in this review, is worthwhile and can be investigated in the future.

## Figures and Tables

**Figure 1 cells-09-00870-f001:**
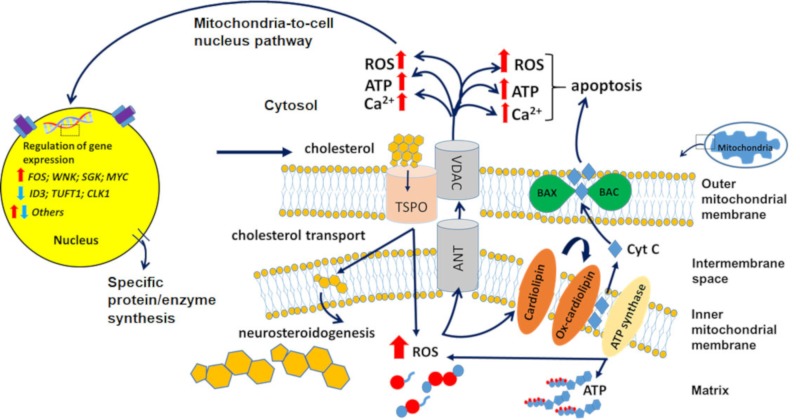
Principal functions of the 18-kDa translocator protein (TSPO): initiation of programmed cell death, including apoptosis, as well as regulation of the cell’s nuclear gene expression. This figure presents an overview of the cellular pathways influenced by TSPO. TSPO favors neurosteroid synthesis by transporting cholesterol over the outer mitochondrial membrane to the inner mitochondrial membrane. Furthermore, it modulates ATP synthase activity, thus initiating reactive oxygen species (ROS) production, which can result in cardiolipin oxidation and opening of the mitochondrial permeability transition pore (mPTP). Opening of the mPTP allows for ATP and Ca^2+^ release, and depolarization of the mitochondrial membrane. This depolarization results in opening of the Bax/Bak channel, allowing for the passage of cytochrome c into the cytosol as an initiating step for the mitochondrial apoptosis cascade. The mitochondrial ROS generation and ATP and Ca^2+^ release are also considered part of the mitochondria-to-nucleus pathway of cell nuclear gene expression modulation, which includes the modulation of hundreds to thousands of genes (see also [8,11,12,13]).

**Figure 2 cells-09-00870-f002:**
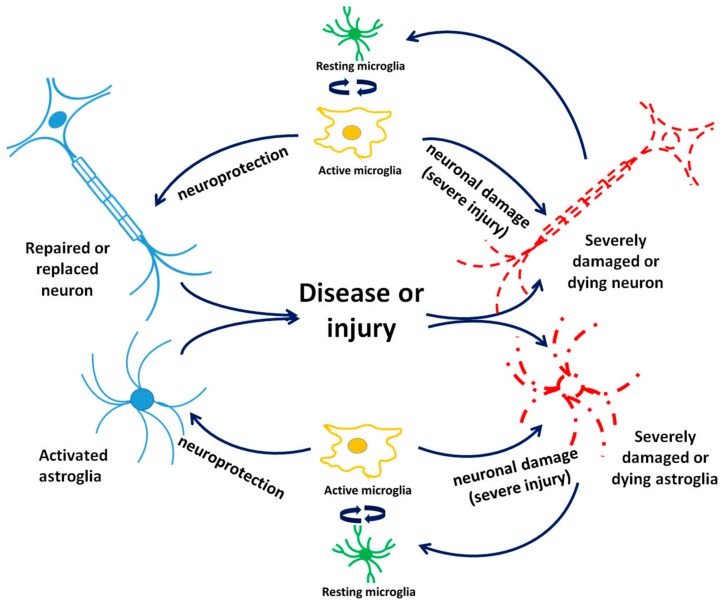
Interactions between brain cell types taking part in acute and progressing brain damage after brain injury and disease. Acute microglial activation following mild brain damage causes may provide protection to neurons and astrocytes while long-term glial activation may cause damage to and the death of neurons and astrocytes.

**Figure 3 cells-09-00870-f003:**
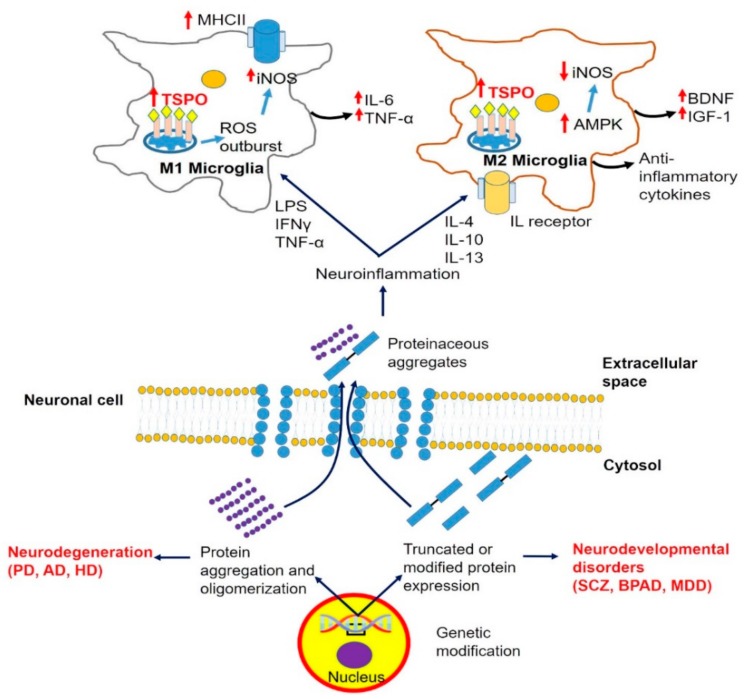
Cellular pathology of neurodegenerative and neurodevelopmental diseases and its correlation with microglial response and modulation of TSPO ligand-binding potentials. Neurodegeneration initiated by cellular protein modification or aggregation induces an inflammatory response, accompanied by activation of the M1 and/or M2 microglial pathway. Proinflammatory (M1) microglial phenotype activated by LPS, Tumor necrosis factor alpha TNF-α, or Interferon gamma IFNγ stimulation is characterized by the expression of inducible nitric oxide synthase iNOS, MHC II, and significantly increased TSPO BP as well as ROS overproduction and secretion of injury-mediated cytokines. The anti-inflammatory (M2) microglial phenotype is stimulated by interleukins (IL-4, IL-10, IL-13) characterized by the secretion of brain-derived neurotrophic factor (BDNF), Insulin growth factor 1 (IGF-1), and anti-inflammatory cytokines [52]. Abbreviations: AD—Alzheimer disease; AMPK—5′ AMP-activated protein kinase; BDNF—brain derived neutrophic factor; BPAD—Bipolar affective disorder; HD—Huntington disease; IFNγ—Interferon gamma; IGF1—Insulin growth factor 1; IL—Interleukin; iNOS—Inducible nitric oxide synthase; LPS—Lipopolysaccharide; MDD—Major depressive disorder; MHCII—Major histocompatibility complex protein class II; PD—Parkinson disease; SCZ—Schizophrenia; TNF-α—Tumor necrosis factor alpha.

**Figure 4 cells-09-00870-f004:**
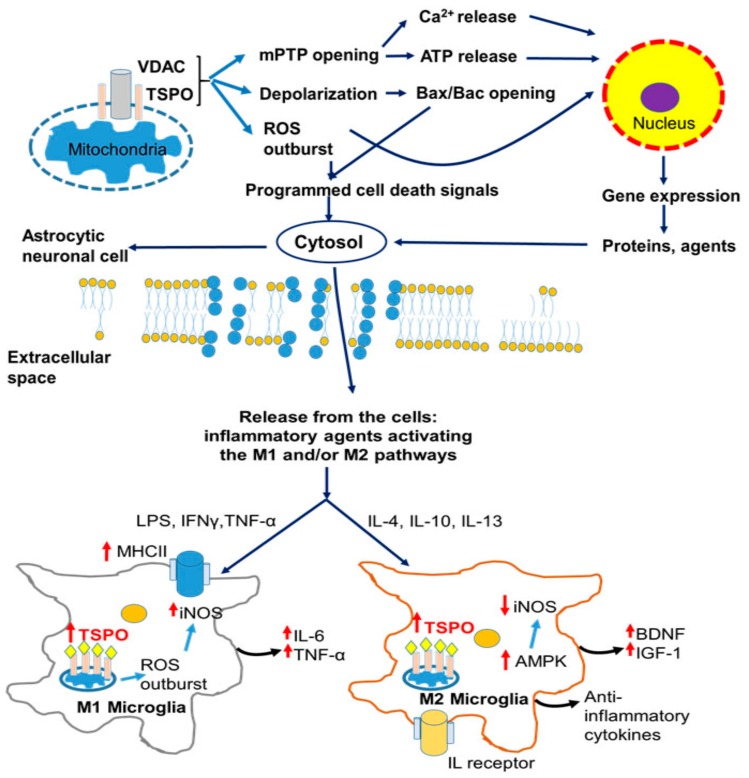
Intracellular manifestation of TBI in neurons and astrocytes, resulting in microglial activation. A vicious cycle of cell death and activation associated with enhanced TSPO responses’ expression in brain injury is reflected by cellular apoptosis and necrosis attained by way of mPTP opening and a decrease of intraneuronal ATP, which consequently triggers the neuroinflammatory response by microglial activation. Proinflammatory (M1) microglial phenotype activated by LPS, TNF-α, or IFNγ stimulation, and characterized by the expression of iNOS, MHCII, and significantly increased TSPO BP, contribute to ROS overproduction and the secretion of injury-mediated cytokines. The anti-inflammatory (M2) microglial phenotype is stimulated by interleukins (IL-4, IL-10, IL-13) and characterized by the secretion of BDNF, IGF-1, and anti-inflammatory cytokines. On the one hand, based on our various studies, we arrive a the assumption that mitochondrial membrane depolarization leads to programmed cell death; on the other hand, less pronounced effects on the mitochondrial membrane potential, but still allowing for ROS generation, and Ca^2+^ and ATP release into the cytosol, will lead to gene expression modulation that is part of: Microglial activation, astrocytic activation, neuronal and astrocytic cell death, neuronal (re) generation, angiogenesis, and wound healing [13]. Abbreviations: AD—Alzheimer’s disease; AMPK-5′—AMP-activated protein kinase; BDNF—brain derived neutrophic factor; BPAD—Bipolar affective disorder; HD—Huntington’s disease; IFNγ—Interferon gamma; IGF1—Insulin growth factor 1; IL—Interleukin; iNOS—Inducible nitric oxide synthase; LPS—Lipopolysaccharide; MDD—Major depressive disorder; MHCII—Major histocompatibility complex protein class II; mPTP—Mitochondrial permeability transition pore; PD—Parkinson’s disease; SCZ—Schizophrenia; TNF-α—Tumor necrosis factor alpha.

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
