# Peer review of "Diagnostic and Therapeutic Potential of TSPO Studies Regarding Neurodegenerative Diseases, Psychiatric Disorders, Alcohol Use Disorders, Traumatic Brain Injury, and Stroke: An Update"

_cells, 2020, doi:10.3390/cells9040870_

Round 1

Reviewer 1 Report

Authors have given a broad overview about translocator protein TSPO, formerly known as peripheral benzodiazepine receptor (PBR), which mainly is localized in the outer mitochondrial membrane. Probably this is why authors describe so much (and well-known)  functions of mitochondria. Although the review  is dedicated to show relationships between TSPO and neurological and psychiatric disorders, the authors frankly  indicate which mechanisms are revealed in this context, which data are contradictory.  Nevertheless the idea about TSPO as potential target for the treatment of brain diseases is worthwhile and can be investigate in the future.

My recommendations: shorten the Part 5 concerning well-known mechanisms of alcohol cellular action, avoid repetition of mitochondrial functions.

After this minor text revision, the manuscript can be recommended for publication.

Author Response

We believe the comments of the reviewers are complementary.  Thus, we think it is prudent to answer all of them in one stretch.  And provide our full response to each one reviewer.

As we also write at the end of our responses :

We truly hope our amendments are in agreement with and alleviate reviewers' concerns.  Just as much we believe these interactions really improved the manuscript at crucial points.  

Reviewer 1

Authors have given a broad overview about translocator protein TSPO, formerly known as peripheral benzodiazepine receptor (PBR), which mainly is localized in the outer mitochondrial membrane.

Response :  Yes, thank you for saying this.

Probably this is why authors describe so much (and well-known)  functions of mitochondria.

Response :  Actually, there is controversy within the relatively small field of TSPO research.  Research of TSPO effects on steroid production  indeed receives a lot of attention.  Whereas other well established functional pathways affected by TSPO as determined by hardworking scientists do not receive their deserved attention.  It occasionally causes vicious contention.  We hope to achieve a more balanced approach to TSPO functions.  Without stoking the fires of contention, which is only counterproductive.  So, please leave it be.

Although the review  is dedicated to show relationships between TSPO and neurological and psychiatric disorders, the authors frankly  indicate which mechanisms are revealed in this context, which data are contradictory. 

Response :  We are very thankful that the reviewer points this out.  Indeed, contradictory results can complicate thinking about what actually happens.  But if and when somebody can spend time on it, the consequent understanding of the underlying causes typically leads to advances in the field. 

Nevertheless the idea about TSPO as potential target for the treatment of brain diseases is worthwhile and can be investigate in the future.

Response :  Here we are also very thankful for the reviewer pointing this out. This actually is the point of the paper.  I myself was kind of surprised  that research of this kind covered a broader array of  brain disorders than I thought a priori.  But it is sporadic.  It needs some more pushing.

My recommendations: shorten the Part 5 concerning well-known mechanisms of alcohol cellular action, avoid repetition of mitochondrial functions.

Response :  If you don't mind, we would like to give the first sentence of the second  reviewer :  " The review article was written with a great flow throughout the manuscript. "

The reason for doing it this way is to reach a broad audience, first of all the people who did the actual experimental research regarding TSPO , to bring them together in one piece of work.  A second major group of course is neuroscientists of various color.  Each one has own specialty.  For each person, what wewrote on their subject of course is not new, and for them not needed.  For all the other people not working on that particular subject it may be helpful.  Hopefully, the review presents a broad enough context for all topics covered to provide a place for each item, in connection with all the other items. ( and yes mitochondrial function is the connector )

After this minor text revision, the manuscript can be recommended for publication.

Response :  we hope that the reviewer sees that all of his comments are taken very seriously.  And as they are expressed, of course they are very important.  We spent time to find a way how to incorporate our thinking, provoked by this reviewer' s comments, in the conclusion. 

Combining our responses  with the conclusions of the reviewer, these two sentences were added to the end of the conclusion.

” We hope that this review presents a sufficient overview of TSPO functions in relation to an organism's varied responses to brain damage and brain disorders.  We believe that the idea about TSPO as potential target for treatment of brain damage, as expressed by the various studies summarized in this review, is worthwhile and can be investigated in the future."

Reviewer 2

The review article was written with a great flow throughout the manuscript.

We very much appreciate this comment, of course.

Few things to be added-

  1. comment on the clarity of PBR28 applications

Response :  PBR28 applications were discussed in lines 243 – 245 and 303 and 358 and in 552-555.

  1. why FDG fails to do the same job

Response :  The reviewer is subtle but cruel.  There are more than 30000 papers on FDG.  Restrict it to brain, you get five thousand, we think. We only have three days left to respond to the reviewer's question. 

FDG primarily assays metabolism (energy).  TSPO as discussed in other papers and reviews functions to maintain homeostasis.  Surely, almost every function modulated to maintain homeostasis requires energy.  But the energy consumption by itself will not say what the function is.  And yes, TSPO ligands as biomarkers have their own drawbacks.  But the following is always true, to combine the TSPO biomarker approach with the FDG biomarker approach, we are truly convinced that will give fruitful results.  If I remember well some studies like that have been done already. 

But at this stage, in this review, the details are beyond the scope of the subject addressed.

  1. TSPO role in COPDs

Response :  Again TSPO  involvement in lung disorders is very interesting, also because brain dysfunction of various kinds apparently can be associated with hampered lung function.  Painfully seen during these days, just as an example, Alzheimer patients are highly vulnerable to pneumonia caused by COVID19. It's very tempting to jump on this bandwagon, but again, it is beyond the scope of this review.

  1. what is the future of TSPO-based biomarkers overall 

Response :  In my opinion TSPO-based biomarkers are and will be primarily useful to study function of TSPO, which indeed is very important. 

This sentence is added to the conclusion (close to its top"Furthermore, TSPO ligands as biomarkers may teach us how TSPO is involved in processes related to brain disorders, including its regulation of various mitochondrial functions."

I am afraid that using them as diagnostics for patients might hamper the treatment, because as TSPO ligands, without going into details, they will strongly affect mitochondrial function. Moreover, the higher the affinity of the ligands for TSPO, the more this will be the case.

But I don't want to inadvertently bring contention into the discussion this in this review. I have seen enough of that.

Reviewer 3

A very well conducted review about the role and potential effects of 18 kDa translocator protein (TSPO) in diagnostic approaches based on nuclear medicine techniques for TBI. The authors also suggest the possible therapeutic pathway and review the , but also therapeutic approaches

Some minor issues:

In the title, authors should remove the semicolon between .. regarding and neurodegenerative .. (Regarding; Neurodegenerative)

Response :  We have done so. (Highlighted in the manuscript)

I think that the abstract is too speculative and does not provide the conclusions of the paper.. i suggest to modify it focusing on the relevance of this paper

Response :  We have made several modifications in the Abstract following the recommendations of the Reviewer 3

The new abstract is copy / pasted here. In the ms. itself, the changes are highlighted.

Abstract: Neuroinflammation and cell death are among the common symptoms of many central nervous system diseases and injuries. Neuroinflammation and programmed cell death of the various cell types in the brain appear to be part of these disorders, and characteristic for each cell type, including neurons and glia cells. Concerning the effects of 18 kDa translocator protein (TSPO) on glial activation, as well as associated with neuronal cell death, as a response mechanism to oxidative stress; the changes of its expression targeted via TSPO specific PET tracers’ uptake could also offer evidence for following the pathogenesis of these disorders, potentially increasing the number of diagnostic tests to accurately establish the stadium and development of the disease in question. Nonetheless, the differences in results regarding TSPO PET signals of first and second generations of tracers measured in patients with neurological disorders versus healthy controls indicate that we still have to understand more regarding TSPO characteristics.. Expanding on investigations regarding the neuroprotective and healing effects of TSPO ligands could also contribute to better understanding of the therapeutic potential of TSPO activity for brain damage due brain injury and disease. Studies so far have directed attention to  the effects on neurons and glia, and processes such as death, inflammation, and regeneration.  It definitely is worthwhile to drive such studies forward.   From recent research it also appears that TSPO ligands such as PK11195, Etifoxine, Emanupil, and 2-Cl-MGV-1 demonstrate the potential of targeting TSPO for treatments of brain diseases and disorders.

To conclude our responses

We truly hope our amendments are in agreement with and alleviate reviewers' concerns.  Just as much we believe these interactions really improved the manuscript at crucial points.                 

Reviewer 2 Report

The review article was written with a great flow throughout the manuscript. Few things to be added-

  1. comment on the clarity of PBR28 applications
  2. why FDG fails to do the same job
  3. TSPO role in COPDs
  4. what is the future of TSPO-based biomarkers overall 

Author Response

We believe the comments of the reviewers are complementary.  Thus, we think it is prudent to answer all of them in one stretch.  And provide our full response to each one reviewer.

Please see response to Reviewer 2 further below.

As we also write at the end of our responses :

We truly hope our amendments are in agreement with and alleviate reviewers' concerns.  Just as much we believe these interactions really improved the manuscript at crucial points.  

Please see response to Reviewer 2 further below.

Reviewer 1

Authors have given a broad overview about translocator protein TSPO, formerly known as peripheral benzodiazepine receptor (PBR), which mainly is localized in the outer mitochondrial membrane.

Response :  Yes, thank you for saying this.

Probably this is why authors describe so much (and well-known)  functions of mitochondria.

Response :  Actually, there is controversy within the relatively small field of TSPO research.  Research of TSPO effects on steroid production  indeed receives a lot of attention.  Whereas other well established functional pathways affected by TSPO as determined by hardworking scientists do not receive their deserved attention.  It occasionally causes vicious contention.  We hope to achieve a more balanced approach to TSPO functions.  Without stoking the fires of contention, which is only counterproductive.  So, please leave it be.

Although the review  is dedicated to show relationships between TSPO and neurological and psychiatric disorders, the authors frankly  indicate which mechanisms are revealed in this context, which data are contradictory. 

Response :  We are very thankful that the reviewer points this out.  Indeed, contradictory results can complicate thinking about what actually happens.  But if and when somebody can spend time on it, the consequent understanding of the underlying causes typically leads to advances in the field. 

Nevertheless the idea about TSPO as potential target for the treatment of brain diseases is worthwhile and can be investigate in the future.

Response :  Here we are also very thankful for the reviewer pointing this out. This actually is the point of the paper.  I myself was kind of surprised  that research of this kind covered a broader array of  brain disorders than I thought a priori.  But it is sporadic.  It needs some more pushing.

My recommendations: shorten the Part 5 concerning well-known mechanisms of alcohol cellular action, avoid repetition of mitochondrial functions.

Response :  If you don't mind, we would like to give the first sentence of the second  reviewer :  " The review article was written with a great flow throughout the manuscript. "

The reason for doing it this way is to reach a broad audience, first of all the people who did the actual experimental research regarding TSPO , to bring them together in one piece of work.  A second major group of course is neuroscientists of various color.  Each one has own specialty.  For each person, what wewrote on their subject of course is not new, and for them not needed.  For all the other people not working on that particular subject it may be helpful.  Hopefully, the review presents a broad enough context for all topics covered to provide a place for each item, in connection with all the other items. ( and yes mitochondrial function is the connector )

After this minor text revision, the manuscript can be recommended for publication.

Response :  we hope that the reviewer sees that all of his comments are taken very seriously.  And as they are expressed, of course they are very important.  We spent time to find a way how to incorporate our thinking, provoked by this reviewer' s comments, in the conclusion. 

Combining our responses  with the conclusions of the reviewer, these two sentences were added to the end of the conclusion.

” We hope that this review presents a sufficient overview of TSPO functions in relation to an organism's varied responses to brain damage and brain disorders.  We believe that the idea about TSPO as potential target for treatment of brain damage, as expressed by the various studies summarized in this review, is worthwhile and can be investigated in the future."

Reviewer 2

The review article was written with a great flow throughout the manuscript.

We very much appreciate this comment, of course.

Few things to be added-

  1. comment on the clarity of PBR28 applications

Response :  PBR28 applications were discussed in lines 243 – 245 and 303 and 358 and in 552-555.

  1. why FDG fails to do the same job

Response :  The reviewer is subtle but cruel.  There are more than 30000 papers on FDG.  Restrict it to brain, you get five thousand, we think. We only have three days left to respond to the reviewer's question. 

FDG primarily assays metabolism (energy).  TSPO as discussed in other papers and reviews functions to maintain homeostasis.  Surely, almost every function modulated to maintain homeostasis requires energy.  But the energy consumption by itself will not say what the function is.  And yes, TSPO ligands as biomarkers have their own drawbacks.  But the following is always true, to combine the TSPO biomarker approach with the FDG biomarker approach, we are truly convinced that will give fruitful results.  If I remember well some studies like that have been done already. 

But at this stage, in this review, the details are beyond the scope of the subject addressed.

  1. TSPO role in COPDs

Response :  Again TSPO  involvement in lung disorders is very interesting, also because brain dysfunction of various kinds apparently can be associated with hampered lung function.  Painfully seen during these days, just as an example, Alzheimer patients are highly vulnerable to pneumonia caused by COVID19. It's very tempting to jump on this bandwagon, but again, it is beyond the scope of this review.

  1. what is the future of TSPO-based biomarkers overall 

Response :  In my opinion TSPO-based biomarkers are and will be primarily useful to study function of TSPO, which indeed is very important. 

This sentence is added to the conclusion (close to its top"Furthermore, TSPO ligands as biomarkers may teach us how TSPO is involved in processes related to brain disorders, including its regulation of various mitochondrial functions."

I am afraid that using them as diagnostics for patients might hamper the treatment, because as TSPO ligands, without going into details, they will strongly affect mitochondrial function. Moreover, the higher the affinity of the ligands for TSPO, the more this will be the case.

But I don't want to inadvertently bring contention into the discussion this in this review. I have seen enough of that.

Reviewer 3

A very well conducted review about the role and potential effects of 18 kDa translocator protein (TSPO) in diagnostic approaches based on nuclear medicine techniques for TBI. The authors also suggest the possible therapeutic pathway and review the , but also therapeutic approaches

Some minor issues:

In the title, authors should remove the semicolon between .. regarding and neurodegenerative .. (Regarding; Neurodegenerative)

Response :  We have done so. (Highlighted in the manuscript)

I think that the abstract is too speculative and does not provide the conclusions of the paper.. i suggest to modify it focusing on the relevance of this paper

Response :  We have made several modifications in the Abstract following the recommendations of the Reviewer 3

The new abstract is copy / pasted here. In the ms. itself, the changes are highlighted.

Abstract: Neuroinflammation and cell death are among the common symptoms of many central nervous system diseases and injuries. Neuroinflammation and programmed cell death of the various cell types in the brain appear to be part of these disorders, and characteristic for each cell type, including neurons and glia cells. Concerning the effects of 18 kDa translocator protein (TSPO) on glial activation, as well as associated with neuronal cell death, as a response mechanism to oxidative stress; the changes of its expression targeted via TSPO specific PET tracers’ uptake could also offer evidence for following the pathogenesis of these disorders, potentially increasing the number of diagnostic tests to accurately establish the stadium and development of the disease in question. Nonetheless, the differences in results regarding TSPO PET signals of first and second generations of tracers measured in patients with neurological disorders versus healthy controls indicate that we still have to understand more regarding TSPO characteristics.. Expanding on investigations regarding the neuroprotective and healing effects of TSPO ligands could also contribute to better understanding of the therapeutic potential of TSPO activity for brain damage due brain injury and disease. Studies so far have directed attention to  the effects on neurons and glia, and processes such as death, inflammation, and regeneration.  It definitely is worthwhile to drive such studies forward.   From recent research it also appears that TSPO ligands such as PK11195, Etifoxine, Emanupil, and 2-Cl-MGV-1 demonstrate the potential of targeting TSPO for treatments of brain diseases and disorders.

To conclude our responses

We truly hope our amendments are in agreement with and alleviate reviewers' concerns.  Just as much we believe these interactions really improved the manuscript at crucial points.                 

Reviewer 3 Report

A very well conducted review about the role and potential effects of 18 kDa translocator protein (TSPO) in diagnostic approaches based on nuclear medicine techniques for TBI. The authors also suggest the possible therapeutic pathway and review the , but also therapeutic approaches

Some minor issues:

In the title, authors should remove the semicolon between .. regarding and neurodegenerative .. (Regarding; Neurodegenerative)

I think that the abstract is too speculative and does not provide the conclusions of the paper.. i suggest to modify it focusing on the relevance of this paper

Author Response

We believe the comments of the reviewers are complementary.  Thus, we think it is prudent to answer all of them in one stretch.  And provide our full response to each one reviewer.

The responses to the comments of reviewer 3, appear below the comments of Reviewers 1 and 2.

As we also write at the end of our responses :

We truly hope our amendments are in agreement with and alleviate reviewers' concerns.  Just as much we believe these interactions really improved the manuscript at crucial points.  

The responses to the comments of reviewer 3, appear below the comments of Reviewers 1 and 2.

Reviewer 1

Authors have given a broad overview about translocator protein TSPO, formerly known as peripheral benzodiazepine receptor (PBR), which mainly is localized in the outer mitochondrial membrane.

Response :  Yes, thank you for saying this.

Probably this is why authors describe so much (and well-known)  functions of mitochondria.

Response :  Actually, there is controversy within the relatively small field of TSPO research.  Research of TSPO effects on steroid production  indeed receives a lot of attention.  Whereas other well established functional pathways affected by TSPO as determined by hardworking scientists do not receive their deserved attention.  It occasionally causes vicious contention.  We hope to achieve a more balanced approach to TSPO functions.  Without stoking the fires of contention, which is only counterproductive.  So, please leave it be.

Although the review  is dedicated to show relationships between TSPO and neurological and psychiatric disorders, the authors frankly  indicate which mechanisms are revealed in this context, which data are contradictory. 

Response :  We are very thankful that the reviewer points this out.  Indeed, contradictory results can complicate thinking about what actually happens.  But if and when somebody can spend time on it, the consequent understanding of the underlying causes typically leads to advances in the field. 

Nevertheless the idea about TSPO as potential target for the treatment of brain diseases is worthwhile and can be investigate in the future.

Response :  Here we are also very thankful for the reviewer pointing this out. This actually is the point of the paper.  I myself was kind of surprised  that research of this kind covered a broader array of  brain disorders than I thought a priori.  But it is sporadic.  It needs some more pushing.

My recommendations: shorten the Part 5 concerning well-known mechanisms of alcohol cellular action, avoid repetition of mitochondrial functions.

Response :  If you don't mind, we would like to give the first sentence of the second  reviewer :  " The review article was written with a great flow throughout the manuscript. "

The reason for doing it this way is to reach a broad audience, first of all the people who did the actual experimental research regarding TSPO , to bring them together in one piece of work.  A second major group of course is neuroscientists of various color.  Each one has own specialty.  For each person, what wewrote on their subject of course is not new, and for them not needed.  For all the other people not working on that particular subject it may be helpful.  Hopefully, the review presents a broad enough context for all topics covered to provide a place for each item, in connection with all the other items. ( and yes mitochondrial function is the connector )

After this minor text revision, the manuscript can be recommended for publication.

Response :  we hope that the reviewer sees that all of his comments are taken very seriously.  And as they are expressed, of course they are very important.  We spent time to find a way how to incorporate our thinking, provoked by this reviewer' s comments, in the conclusion. 

Combining our responses  with the conclusions of the reviewer, these two sentences were added to the end of the conclusion.

” We hope that this review presents a sufficient overview of TSPO functions in relation to an organism's varied responses to brain damage and brain disorders.  We believe that the idea about TSPO as potential target for treatment of brain damage, as expressed by the various studies summarized in this review, is worthwhile and can be investigated in the future."

Reviewer 2

The review article was written with a great flow throughout the manuscript.

We very much appreciate this comment, of course.

Few things to be added-

  1. comment on the clarity of PBR28 applications

Response :  PBR28 applications were discussed in lines 243 – 245 and 303 and 358 and in 552-555.

  1. why FDG fails to do the same job

Response :  The reviewer is subtle but cruel.  There are more than 30000 papers on FDG.  Restrict it to brain, you get five thousand, we think. We only have three days left to respond to the reviewer's question. 

FDG primarily assays metabolism (energy).  TSPO as discussed in other papers and reviews functions to maintain homeostasis.  Surely, almost every function modulated to maintain homeostasis requires energy.  But the energy consumption by itself will not say what the function is.  And yes, TSPO ligands as biomarkers have their own drawbacks.  But the following is always true, to combine the TSPO biomarker approach with the FDG biomarker approach, we are truly convinced that will give fruitful results.  If I remember well some studies like that have been done already. 

But at this stage, in this review, the details are beyond the scope of the subject addressed.

  1. TSPO role in COPDs

Response :  Again TSPO  involvement in lung disorders is very interesting, also because brain dysfunction of various kinds apparently can be associated with hampered lung function.  Painfully seen during these days, just as an example, Alzheimer patients are highly vulnerable to pneumonia caused by COVID19. It's very tempting to jump on this bandwagon, but again, it is beyond the scope of this review.

  1. what is the future of TSPO-based biomarkers overall 

Response :  In my opinion TSPO-based biomarkers are and will be primarily useful to study function of TSPO, which indeed is very important. 

This sentence is added to the conclusion (close to its top"Furthermore, TSPO ligands as biomarkers may teach us how TSPO is involved in processes related to brain disorders, including its regulation of various mitochondrial functions."

I am afraid that using them as diagnostics for patients might hamper the treatment, because as TSPO ligands, without going into details, they will strongly affect mitochondrial function. Moreover, the higher the affinity of the ligands for TSPO, the more this will be the case.

But I don't want to inadvertently bring contention into the discussion this in this review. I have seen enough of that.

Reviewer 3

A very well conducted review about the role and potential effects of 18 kDa translocator protein (TSPO) in diagnostic approaches based on nuclear medicine techniques for TBI. The authors also suggest the possible therapeutic pathway and review the , but also therapeutic approaches

Some minor issues:

In the title, authors should remove the semicolon between .. regarding and neurodegenerative .. (Regarding; Neurodegenerative)

Response :  We have done so. (Highlighted in the manuscript)

I think that the abstract is too speculative and does not provide the conclusions of the paper.. i suggest to modify it focusing on the relevance of this paper

Response :  We have made several modifications in the Abstract following the recommendations of the Reviewer 3

The new abstract is copy / pasted here. In the ms. itself, the changes are highlighted.

Abstract: Neuroinflammation and cell death are among the common symptoms of many central nervous system diseases and injuries. Neuroinflammation and programmed cell death of the various cell types in the brain appear to be part of these disorders, and characteristic for each cell type, including neurons and glia cells. Concerning the effects of 18 kDa translocator protein (TSPO) on glial activation, as well as associated with neuronal cell death, as a response mechanism to oxidative stress; the changes of its expression targeted via TSPO specific PET tracers’ uptake could also offer evidence for following the pathogenesis of these disorders, potentially increasing the number of diagnostic tests to accurately establish the stadium and development of the disease in question. Nonetheless, the differences in results regarding TSPO PET signals of first and second generations of tracers measured in patients with neurological disorders versus healthy controls indicate that we still have to understand more regarding TSPO characteristics.. Expanding on investigations regarding the neuroprotective and healing effects of TSPO ligands could also contribute to better understanding of the therapeutic potential of TSPO activity for brain damage due brain injury and disease. Studies so far have directed attention to  the effects on neurons and glia, and processes such as death, inflammation, and regeneration.  It definitely is worthwhile to drive such studies forward.   From recent research it also appears that TSPO ligands such as PK11195, Etifoxine, Emanupil, and 2-Cl-MGV-1 demonstrate the potential of targeting TSPO for treatments of brain diseases and disorders.

To conclude our responses

We truly hope our amendments are in agreement with and alleviate reviewers' concerns.  Just as much we believe these interactions really improved the manuscript at crucial points.